# Feasibility and Activity of Megestrol Acetate in Addition to Etoposide, Doxorubicin, Cisplatin, and Mitotane as First-Line Therapy in Patients with Metastatic/Unresectable Adrenocortical Carcinoma with Low Performance Status

**DOI:** 10.3390/cancers15184491

**Published:** 2023-09-09

**Authors:** Antonella Turla, Marta Laganà, Andrea Abate, Valentina Cremaschi, Manuel Zamparini, Matteo Chittò, Francesca Consoli, Andrea Alberti, Roberta Ambrosini, Mariangela Tamburello, Salvatore Grisanti, Guido Alberto Massimo Tiberio, Sandra Sigala, Deborah Cosentini, Alfredo Berruti

**Affiliations:** 1Medical Oncology Unit, ASST Spedali Civili, Department of Medical and Surgical Specialties, Radiological Sciences and Public Health, University of Brescia, Piazzale Spedali Civili 1, 25123 Brescia, Italy; a.turla@unibs.it (A.T.); v.cremaschi@unibs.it (V.C.); manuel.zamparini@unibs.it (M.Z.); m.chitto@studenti.unibs.it (M.C.); francesca.consoli@libero.it (F.C.); a.alberti015@unibs.it (A.A.); salvatore.grisanti@unibs.it (S.G.); d.cosentini@unibs.it (D.C.); alfredo.berruti@unibs.it (A.B.); 2Section of Pharmacology, Department of Molecular and Translational Medicine, University of Brescia, Viale Europa 11, 25123 Brescia, Italy; andrea.abate@unibs.it (A.A.); mariangela.tamburello@unibs.it (M.T.); sandra.sigala@unibs.it (S.S.); 3Radiology Unit, ASST Spedali Civili, Piazzale Spedali Civili 1, 25123 Brescia, Italy; robertambrosini@gmail.com; 4Surgical Unit, ASST Spedali Civili, Department of Medical and Surgical Specialties, Radiological Sciences, and Public Health, University of Brescia, Piazzale Spedali Civili 1, 25123 Brescia, Italy; guido.tiberio@unibs.it

**Keywords:** adrenocortical carcinoma, megestrol acetate, chemotherapy

## Abstract

**Simple Summary:**

Preclinical studies have shown an anti-neoplastic effect of progestins against adrenal cortical carcinoma. Progestins have a positive effect on patient cenesthesia and may make standard chemotherapy more tolerable. In this study, the addition of megestrol acetate to the etoposide, doxorubicin, cisplatin, and mitotane regimen (EDP-M) in patients with ACC and a low PS allowed the administration of EDP-M at adequate doses, and the efficacy was not inferior to that of EDP-M administered to patients with a good PS. Since a low PS is a predictive factor of poor efficacy of antineoplastic treatments in cancer patients, these results suggest a potentiating effect of megestrol acetate to EDP-M and provide the rationale for the ongoing randomized study at the Medical Oncology of Brescia.

**Abstract:**

(1) Background: The standard first-line therapy for advanced adrenocortical carcinoma (ACC) is represented by EDP-M (etoposide, doxorubicin, cisplatin + mitotane). Progestins have shown cytotoxic activity both in vitro and in vivo on ACC; better EDP-M tolerability and efficacy have been hypnotized due to the association with progestins. (2) Methods: The feasibility and tolerability of EDP-M combined with oral megestrol acetate (EDP-MM) were tested in 24 patients (pts) affected by metastatic ACC with a low performance status (PS); the case group was compared with a 48 pts control group according to the propensity score. The secondary objectives were clinical benefit rate (CBR), progression-free survival (PFS), and overall survival (OS). (3) Results: Thirteen pts (54.2%) in the EDP-MM population experienced progestin-related toxicities; in particular, five pts experienced vaginal bleeding (20.8%); four pts experienced weight gain (16.7%); and thromboembolic events, worsening of hypertension, skin rashes, and hyperglycemia were registered in one patient each (4.2%). This led to the discontinuation of megestrol acetate in four pts (16.7%). EDP-M-related toxicities were similar in both groups. No differences in PFS and OS curves were observed; the CBR was 75.0% and 60.4%, respectively. (4) Conclusions: The association of EDP-M + megestrol acetate in ACC pts with a low PS is feasible and well tolerated; its efficacy appeared to be non-inferior to EDP-M administered to pts with a good PS.

## 1. Introduction

Adrenocortical carcinoma (ACC) is a rare and aggressive malignancy [1], the management of which is challenging [2]. The EDP-M scheme (etoposide, doxorubicin, cisplatin, and mitotane) is the standard systemic treatment of advanced ACC [3]. The results of a prospective multicenter randomized FIRM-ACT trial reported a significant advantage in terms of either progression-free survival (PFS) or overall survival (OS) of EDP-M compared to Streptozotocin plus mitotane (Sz-M) [4]. Although the EDP-M regimen can induce complete pathological remissions in a small proportion of patients, its overall efficacy is limited and the prognosis of patients with metastatic disease remains poor, with less than 15% surviving patients at 5 years [2,4,5]. Second-line chemotherapies and target therapies tested up to now have very limited or no efficacy [6,7,8,9,10,11]. Immunotherapy has achieved long-term control in a small proportion of patients with metastatic ACC [12,13,14,15,16]. Although the results of immunotherapy are promising, they are, by far, inferior to those obtained in other tumors in which this strategy is a standard therapy, highlighting the need to identify patients destined to benefit from this therapy and to implement strategies to overcome the intrinsic immunoresistance of ACC [17]. In this scenario, another line of research is needed to enhance the efficacy of EDP-M.

Progesterone (Pg) has been demonstrated to act as an anti-tumoral drug in different cancers [18]. Preclinical evidence from our group provided support for the role of this hormone as an anticancer drug also in ACC, providing the basis for the use of progesterone in clinics. Originally, we demonstrated that the antineoplastic activity of abiraterone acetate is mediated by a drug-induced increase in progesterone levels. Progesterone in turn was found to reduce cell viability in the ACC cell line NCI-H295R cells and in primary secreting ACC cell cultures through its receptors (PgRs) [19], involving both genomic and non-genomic pathways [20,21]. Progesterone also reduced β-catenin nuclear translocation, a pathway frequently altered in ACC, and enhanced the antineoplastic activity of mitotane [20] and the CDK4/6 inhibitor ribociclib [22]. Metastasis-derived cell models, namely MUC-1 and TVBF-7 cell lines, were found to be less sensitive to the Pg effect due to weaker PgR expression compared to NCI-H295R cells [23]. Interestingly, the cytotoxic effect of progesterone, when present, is maintained after drug withdrawal [24].

More recently, the antineoplastic effect of progesterone on the above-cited cell lines was confirmed in an in vivo zebrafish embryo xenograft model. In this model, progesterone was also found to inhibit metastasis formation in tumor xenografts derived from MUC-1 and TVBF-7 metastatic cells, confirming the in vitro results where migration and invasion abilities were significantly suppressed by this hormone [24].

In the past, progestins have represented a therapeutic strategy in the management of patients with breast and endometrial tumors [25,26]. Today, the therapeutic effects of progestins for improving appetite and promoting weight gain are currently exploited to treat cancer-induced weight loss and anorexia [27,28]. A Cochrane systematic review revealed in fact a significant benefit for appetite and weight gain in oncologic patients treated with megestrol acetate (a progestin compound) for at least eight weeks [29]. In relation to the beneficial effect of progestins on patients’ cenesthesia and considering the remarkable toxicity of EDP-M in terms of asthenia, nausea, and hypo/anorexia, there is a rationale to introduce megestrol acetate in association with EDP-M in the management of advanced ACC patients with a low performance status in order to make the cytotoxic regimen more tolerable in this patient subgroup [30].

In this paper, we report the results regarding the feasibility and tolerability of megestrol acetate in association with EDP-M in advanced ACC patients with impaired PS. As a secondary aim, we evaluated the efficacy of the EDP-M and megestrol (EDP-MM) combination regimen. 

## 2. Patients and Methods

### 2.1. Patients

Twenty-four patients with advanced/metastatic ACC followed at the Medical Oncology Unit, ASST Spedali Civili Brescia from 1 January 2012 to 1 November 2022 were included in the present study. To be included, patients had to meet the following inclusion criteria: age of 18 years or older; histologically proven ACC; locally advanced or metastatic disease not amenable to surgery; no previous systemic therapies for advanced disease, except for single agent mitotane; a moderately compromised clinical condition in terms of asthenia; inappetence/anorexia and/or ECOG PS 1 or 2; the ability to understand the treatment procedures; and signed informed consent. The exclusion criterion was a previous history of other cancers diagnosed in the previous ten years, except for basal cell or squamous cell carcinoma of the skin or in situ carcinoma of the uterine cervix. 

Patients were treated with the standard EDP-M regimen consisting of etoposide 100 mg/m^2^ on days 2–4, doxorubicin 40 mg/m^2^ on day 1, cisplatin 40 mg/m^2^ on days 3–4, and mitotane, administered in combination with megestrol acetate (320 mg daily). The administration of megestrol acetate continued throughout the period of chemotherapy administration.

A control group of patients, followed at the Medical Oncology Unit of ASST-Spedali Civili, with the same disease characteristics but with general good condition submitted to first-line EDP-M was selected according to the propensity score procedure to compare activity and tolerability of the EDP-MM regimen to the EDP-M scheme. The inclusion criteria for the control group include unresectable/metastatic ACC; absence of other diseases, especially renal, cardiological, and/or neurological comorbidities (for example, congestive heart failure or renal impairment); previous treatment with megestrol acetate or other antineoplastic drugs for ACC, except for mitotane; and to have provided the written informed consent.

### 2.2. Methods 

All data were obtained by reviewing patient history, medical records, and source documents. The data collected were clinical and demographical characteristics, the date and type of surgery, stage at diagnosis, pathology reports (Weiss score and Ki-67 index), hormonal status, date of start and stop of adjuvant mitotane treatment, date of recurrence and type of recurrence (single or multiple, local, or distant), and the date of last follow-up or death. The tumor stage was established according to the ENSAT classification (I, confined tumors ≤ 5 cm; II, confined tumors > 5 cm; III, positive lymph nodes or infiltrating neighboring organs/veins without distant metastases; IV, distant metastases). 

The primary objective of the study was the evaluation of the feasibility and tolerability of megestrol acetate in association with the EDP-M chemotherapy regimen as compared to EDP-M alone. The secondary objective was the comparison of the efficacy of EDP-MM versus EDP-M treatment in terms of the Clinical Benefit Rate (CBR); progression-free survival (PFS), defined as the time from medical or surgical treatment to the progression of disease or death from any cause; and overall survival (OS), defined as the time from diagnosis to patient death or the date of the last follow-up.

This retrospective study was approved by the Ethical Review Board of ASST-Spedali Civili in Brescia (Protocol number: 5525). Written informed consent was obtained from each patient.

### 2.3. Statistical Analysis

#### 2.3.1. Study Power

According to the results of the FIRM-ACT trial [4], patients with locally advanced or metastatic ACC who received treatment with EDP-M experienced serious toxicities in 60% of cases. Assuming that megestrol acetate will maintain this proportion, we can estimate the required sample size needed to achieve a certain level of precision or margin of error. To calculate the sample size, we can use the formula $N=\frac{{Z^{2}}_{\alpha /2}\cdot p\cdot (1-p)}{{\left(MoE\right)}^{2}}$, where *N* is the sample size, *Z* is the Z-score corresponding to the desired level of confidence, *p* is the estimated proportion of toxicities, and *MoE* is the desired margin of error.

Assuming a type I error of 5%, the calculated margin of error for the estimate of the toxicity proportion in patients treated with EDP-MM is between 18% and 22%. Therefore, based on this margin of error, we would need to treat at least 24 patients with EDP-MM to achieve the desired level of precision in the estimate of toxicity rates.

#### 2.3.2. Collection and Statistical Analysis of Data

The 24 cases and 48 controls were matched 1:2 according to the available factors of the GRAS score [5] (which includes an age greater or less than 50 years and presence/absence of symptoms) using propensity score matching (PSM). The clinical and anatomopathological variables were computed as categorical or continuous variables and described through frequency tables.

The primary endpoint was expressed as the proportion of subjects, toxic events, and treatment and was reported in contingency tables. All statistical analyses were conducted in SPSS software version 23.0 (SPSS Inc., Chicago, IL, USA), considering a 5% significance threshold. Comparisons between the different groups were performed with the chi-square test and *t*-test for independent samples, respectively, for categorical and continuous variables. Survival functions were computed using the Kaplan–Meier method and compared using a log-rank test. 

## 3. Results

### 3.1. Patient Characteristics

Table 1 shows the clinical–pathological characteristics of the 24 patients who received EDP-MM (cases) and the 48 patients who were submitted to EDP-M alone (controls).

Cases were mostly women (87.5% vs. 62.5%) (*p* 0.028); the median age was similar at 46 years in both groups.

At baseline, before starting EDP-M, the group of cases presented the following clinical features: 13 patients (54.2%) complained of mass symptoms, 6 (25%) hormonal symptoms, and 4 (16.6%) both mass symptoms and hormonal symptoms. Hormonal symptoms in the control group were observed in 25 patients (52.1%), mass symptoms in 14 (29.2%), and both mass and hormonal symptoms in 4 (8.3%). ACC was hormone-secreting in 18 (94.7%) EDP-MM versus 27 (62.8%) EDP-M patients, respectively (*p* 0.012).

An ECOG Performance Status (PS) of > 1 was observed in 70.8% of EDP-MM-treated patients vs. 14.6% of EDP-M ones (*p* < 0.001).

The stage at diagnosis in the EDP-MM group was distributed as follows: 20.8% stage II, 20.8% stage III, and 58.4% stage IV. The corresponding distribution in the EDP-M group was 27.7% stage I–II, 34.0% stage III, and 38.3% stage IV (*p* 0.267). The GRAS score at diagnosis was mostly pejorative in both groups (77.8% and 62.3%, respectively). 

Most patients in both groups underwent surgery as a first treatment (75% and 72.9% in the cases and controls, respectively, *p* 0.85) which was radical (R0) in 55.5% of cases versus 74.3% of controls (*p* 0.096). 

At the last follow-up examination in November 2022, 10 (41.7%) patients of the case group and 13 (27.1%) patients of the control group were alive.

### 3.2. Treatment-Related Toxicities

The median duration of megestrol acetate administration was 5.5 months (range 2–19 months).

Megestrol-acetate-related toxicities were observed in 13 out of 24 patients (54.2%). They were vaginal bleeding in five patients (20.8%) and weight gain in four patients (16.7%), while thromboembolic events, worsening hypertension, skin rash, and hyperglycemia were observed in one patient each (4.2%). Side effects led to megestrol acetate interruption in four patients (16.7%). The causes of drug withdrawal were vaginal bleeding in three patients and skin toxicity in the remaining patient.

Table 2 shows the toxicities related to EDP-M treatment in all cases and in 39 controls (for whom toxicity data were available). Twenty-three patients (95.8%) receiving megestrol acetate developed at least one EDP-M-related toxicity. In the control group, side effects occurred in 38 patients (97.4%) (*p* 0.725). Seventeen patients treated with megestrol acetate (70.9%) developed nausea, while this symptom was reported in fourteen patients treated with EDP-M (35.9%) (*p* 0.010). Conversely, no difference in terms of vomiting was observed between the two groups: 33.3% (8 patients) of the case group versus 46.2% (18 patients) of the control group (*p* 0.315). Diarrhea occurred in only one patient in the control group (2.6%) and no patient in the megestrol acetate group (*p* 0.42). Asthenia was noted in 20 patients (83.3%) of the case group, versus 23 patients (59%) of the control cohort (*p* 0.044). Constipation was reported in 16.7% and 7.7% of the case and control groups, respectively (*p* 0.271). Hematological toxicities were frequent both in the cases (50%) and controls (33%) (*p* 0.189). Anemia was observed in 12 EDP-MM patients (50%) and 11 (28.2%) EDP-M ones (*p* 0.108). Neutropenia was observed in four (16.7%) and three (7.7%) patients in the case and control groups, respectively (*p* 0.271). Thrombocytopenia was observed in five (20.8%) and two (5.1%) patients, respectively (*p* 0.095). Other toxicities, such as neurological, hypoadrenalism, cardiac toxicity, and hepatic toxicity, were reported in the EDP-M group only.

Grade 3–4 toxicities were observed in 44.3% of cases in the whole series and they were similarly distributed in both groups (Table 2). No grade 5 adverse events were observed.

Seven EDP-MM patients’ (29.2%) and ten EDP-M patients’ (20.8%) EDP doses were reduced for treatment-related toxicities.

The median duration of mitotane treatment was 22.45 months (4–62) and 15.63 (1–152) for the EDP-MM and EDP-M patients, respectively (*p* 0.640). Mitotane blood plasma levels between 14 and 20 mg/L were observed in 58.3% and 52.9% of the megestrol and control groups, respectively (*p* 0.684). The median dose of cortisone acetate supplementation was 50 mg daily in both populations (range of 25–100 mg for the EDP-MM patients and 25–75 mg for the EDP-M patients, *p* 0.381).

### 3.3. Clinical Benefit Rate 

Twelve patients (50.0%) in the EDP-MM and twenty patients (41.7%) in the EDP-M group attained a complete + partial radiological response, six (25.0%) and nine (18.7%) experienced disease stabilization, and six (25.0%) and nineteen (39.6%) experienced disease progression (Table 3). Therefore, the proportion of patients attaining a clinical benefit, defined as the percentage of patients who achieved a complete response, partial response, or stable disease to treatment, was 75.0% vs. 60.4%, respectively (*p* = 0.266).

### 3.4. Efficacy of EDP-M and Megestrol Treatment in Terms of PFS and OS

After a median follow-up of 20.13 months, the Kaplan–Meier estimates of median PFS was 8.83 months (6.45–11.28) in the EDP-MM patient group and 8.23 months (4.85–11.60) in the EDP-M one, *p* 0.798 (Figure 1). The corresponding median OS was 27.53 (16.83–38.23) and 29.06 (23.49–34.63) months in the two groups, respectively, *p* 0.777 (Figure 2).

## 4. Discussion

The current standard systemic therapy for advanced/metastatic ACC still remains EDP-M. This regimen is associated with relevant toxicity and tolerance in patients with a low-performance status represents an unsolved issue. Since progestins are efficacious in counteracting cancer-associated anorexia and cachexia, we administered megestrol acetate in association with EDP-M in ACC patients with compromised PS with the aim to improve the patient’s general conditions and mitigate some side effects of EDP-M. In addition, based on preclinical data from our group showing a specific antineoplastic activity of progesterone in in vitro and in vivo ACC experimental models [19,20,23,24] and a synergistic effect when combined with other cytotoxic therapies, we also explored the impact of the addition of megestrol acetate on the safety of the EDP-M regimen. It should be underlined, however, that in in vitro experimental models, synthetic progestin megestrol acetate is able to bind and activate other steroid hormone receptors, including glucocorticoid receptors [31,32,33]. Thus, the contribution of these steroid receptors besides PgRs in modulating the comprehensive effect of megestrol acetate in patients cannot be completely excluded. 

The results showed that the combination of megestrol acetate with EDP-M is feasible. Megestrol acetate was associated with additional toxicities which are typical of this drug, such as weight gain (which was indeed a desirable effect in this patient setting), vaginal bleeding, hyperglycemia, and thromboembolism. However, these side effects were manageable and less than 20% of patients interrupted treatment due to toxicity. With regard to EDP-M-specific toxicities, we compared the frequency and severity of EDP-M side effects of the group that also received megestrol acetate to a control group matched with the propensity score. Interestingly, no difference in toxicity was observed between the two groups, indicating that the addition of megestrol acetate improved the tolerability of EDP-M in low PS patients, allowing them access to treatment. Only nausea was more frequent in the EDP-MM group as compared to EDP-M one, this difference could be mainly attributable to the difference in terms of general conditions between the two groups of patients instead of an additional effect due to the progestins. Altogether, these results demonstrate the feasibility of the EDP-M + megestrol acetate association. Noteworthy, the dose intensity of the EDP regimen was similar in the two groups and the proportion of cycles administered at reduced doses was comparable in the two groups. These data suggest that the addition of megestrol acetate to chemotherapy could allow ACC patients with a low PS to receive EDP-M at adequate doses and dose intensities.

Regarding efficacy, we did not observe any difference in terms of disease response, clinical benefit, PFS, and OS between the two treatment groups. Therefore, the EDP-MM combination administered to patients with a low PS was not inferior to EDP-M administered to patients with a good PS and this is an interesting finding; however, based on these results, we cannot confirm the potentiating effect of megestrol acetate on the efficacy of EDP-M observed in vitro. Cancer treatments are notoriously less effective in patients with a poor PS and the expected prognosis of these patients is lower than that of those with a good PS. Therefore, the observation of a similar prognosis of EDP-MM-treated patients with a poor PS compared to EDP-M patients with a good PS indirectly suggests that the addition of progestin could improve the efficacy of EDP-M. The efficacy of EDP-MM versus EDP is currently being tested in a prospective randomized study underway at our center (PESETA study, EudraCT no 2020-004530-38).

The strength of this paper is that this is the first clinical study that has tested the activity of megestrol acetate associated with EDP-M in patients with ACC; the retrospective design and the lack of some prognostic information in a number of patients in the two matched groups are the major limitations.

## 5. Conclusions

The results of this study show that the combination of EDP-M and megestrol acetate is feasible in ACC patients with a low PS and, thanks to its positive effect on the patient’s cenesthesia, megestrol acetate could allow the administration of the EDP-M regimen at a dose intensity comparable to that administered to patients with a good PS. The non-inferiority in terms of the efficacy of EDP-MM administered to patients with a poor PS compared to EDP-M administered to patients with a good PS does not exclude a potential synergistic effect of progestogen added to standard chemotherapy treatment in patients with ACC.

## Figures and Tables

**Figure 1 cancers-15-04491-f001:**
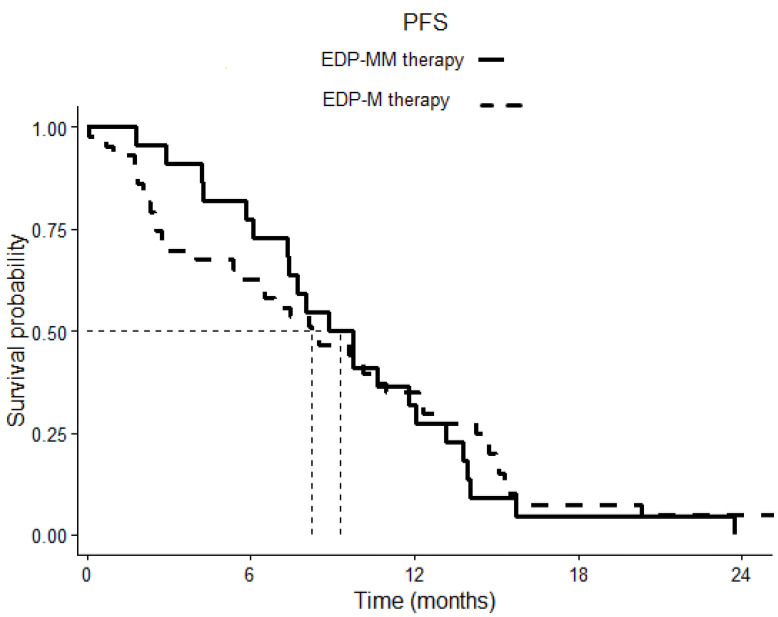
Comparison of progression-free survival (PFS) in patients treated with EDP-M plus megestrol acetate versus patients treated with EDP-M alone. The continuous line indicates cases, the dotted line indicates controls. *p* = 0.798.

**Figure 2 cancers-15-04491-f002:**
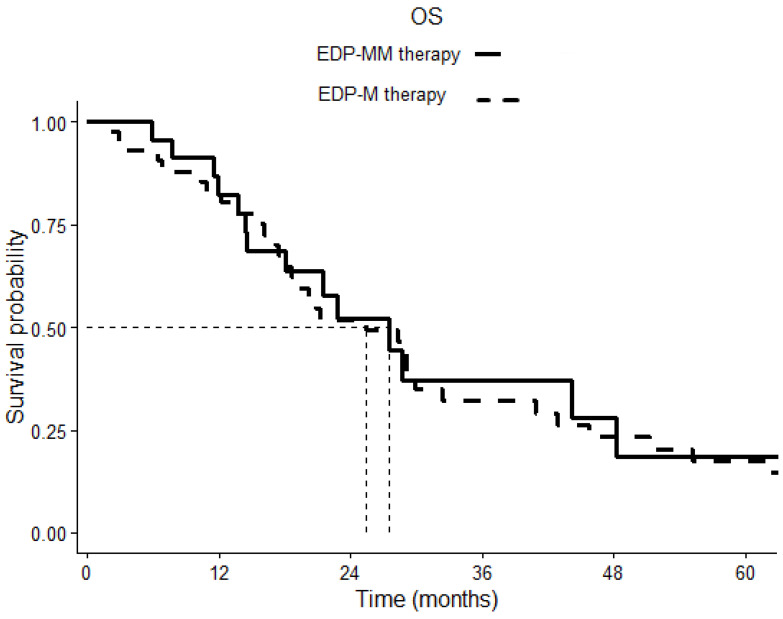
Overall survival (OS) comparison of patients treated with EDP-M plus megestrol acetate and patients treated with EDP-M chemotherapy alone. The continuous line indicates cases, the dotted line indicates controls. *p* = 0.777.

**Table 1 cancers-15-04491-t001:** Patient characteristics of the two matched groups.

Patient Characteristics	EDP-MM1 Patients	EDP-M^2^ Patients	*p*
**Total patients**	24	48	-
**Gender**			**0.028**
Male	3 (12.5%)	18 (37.5%)	
Female	21 (87.5%)	30 (62.5%)	
Missing	0	0	
**Age at diagnosis**			0.892
Median	46.48	46	
Range	19–67	16–72	
Missing	0	0	
**Clinical presentation**			0.601
Mass symptoms	13 (54.2%)	25 (52.1%)	
Hormonal symptoms	6 (25%)	14 (29.2%)	
Mass/hormonal symptoms	4 (16.6%)	4 (8.3%)	
Incidentaloma	1 (4.2%)	5 (10.4%)	
Missing	0	0	
**Secreting tumor** *			**0.012**
Yes	18 (94.7%)	27 (62.8%)	
No	1 (5.3%)	16 (37.2%)	
Missing	5	2	
**ENSAT stage at diagnosis** *			0.267
1–2	5 (20.8%)	13 (27.7%)	
3	5 (20.8%)	16 (34.0%)	
4	14 (58.4%)	18 (38.3%)	
Missing	0	1	
**GRAS at diagnosis** *			0.408
Favorable	0	2 (4.4%)	
Unfavorable	4 (22.2%)	15 (33.3%)	
Pejorative	14 (77.8%)	28 (62.3%)	
Missing	6	3	
**Surgery**			0.850
Yes	18 (75%)	35 (72.9%)	
No	6 (25%)	13 (27.1%)	
Missing	0	0	
**Radical surgery**			0.096
R0	10 (55.5%)	26 (74.3%)	
R1-R2	8 (44.4%)	9 (25.7%)	
Missing	0	0	
**Mitotane in an adjuvant setting**			0.085
Yes	4 (22.2%)	15 (42.9%)	
No	14 (77.8%)	20 (57.1%)	
Missing	0	0	

* The common denominator of the proportions refers to the number of patients for whom data are available. EDP-MM^1^: etoposide 100 mg/m^2^ days 2–4, doxorubicin 40 mg/m^2^ day 1, cisplatin 40 mg/m^2^ days 3–4 + mitotane + megestrol acetate 320 mg daily. EDP-M^2^: etoposide 100 mg/m^2^ days 2–4, doxorubicin 40 mg/m^2^ day 1, cisplatin 40 mg/m^2^ days 3–4 + mitotane.

**Table 2 cancers-15-04491-t002:** Comparison of the toxicities highlighted in the case group and the control group.

Toxicity Related to EDP-M^2^ Treatment	EDP-MM1 Patients			EDP-M^2^ Patients			*p* **
**Treatment-related toxicities** *							
Yes	23 (95.8%)			38 (97.4%)			
No	1 (4.2%)			1 (2.6%)			0.725
Unavailable	0			9			
	**Any grade**	**G1–2**	**G3–4**	**Any grade**	**G1–2**	**G3–4**	
Nausea	17 (70.9%)	16 (66.7%)	1 (4.2%)	14 (35.9%)	12 (30.8%)	2 (5.1%)	**0.010**
Vomiting	8 (33.3%)	8 (33.3%)	0	18 (46.2%)	16 (41.1%)	2 (5.1%)	0.315
Diarrhea	0	0	0	1 (2.6%)	1 (2.6%)	0	0.421
Asthenia	20 (83.3%)	17 (70.8%)	3 (12.5%)	23 (59%)	20 (51.3%)	3 (7.7%)	**0.044**
Constipation	4 (16.7%)	4 (16.7%)	0	3 (7.7%)	2 (5.1%)	1 (2.6%)	0.271
Hematological	12 (50%)	10 (41.7%)	2 (8.3%)	13 (33.3%)	11 (28.2%)	2 (5.1%)	0.189
Neutropenia	4 (16.6%)	2 (8.3%)	2 (8.3%)	3 (7.7%)	2 (5.1%)	1 (2.6%)	0.271
Thrombocytopenia	5 (20.8%)	3 (12.5%)	2 (8.3%)	2 (5.1%)	1 (2.5%)	1 (2.5%)	0.095
Anemia	12 (50%)	10 (41.7%)	2 (8.3%)	11 (28.2%)	9 (20.5%)	3 (7.7%)	0.108
Other toxicities	0	0	0	14 (35.9%)	14 (35.9%)	0	**0.001**

* The common denominator of the proportions refers to the number of patients for whom data are available. ** The *p*-value refers to the comparison between the “any grade” categories. EDP-MM^1^: etoposide 100 mg/m^2^ days 2–4, doxorubicin 40 mg/m^2^ day 1, cisplatin 40 mg/m^2^ days 3–4 + mitotane + megestrol acetate 320 mg daily. EDP-M^2^: etoposide 100 mg/m^2^ days 2–4, doxorubicin 40 mg/m^2^ day 1, cisplatin 40 mg/m^2^ days 3–4 + mitotane.

**Table 3 cancers-15-04491-t003:** Treatment activity in the two groups.

Response to Systemic Treatment	EDP-MM^1^ Patients	EDP-M^2^ Patients	*p*
CR + PR + SD * CR PR SD	18 (75%) 0 12 (50%) 6 (25%)	29 (60.4%) 1 (2.1%) 19 (39.6%) 9 (18.7%)	0.224
PD **	6 (25%)	19 (39.6%)	

* CR: complete response; PR: partial response; SD: stable disease. ** PD: progressive disease. EDP-MM^1^: etoposide 100 mg/m^2^ days 2–4, doxorubicin 40 mg/m^2^ day 1, cisplatin 40 mg/m^2^ days 3–4 + mitotane + megestrol acetate 320 mg daily. EDP-M^2^: etoposide 100 mg/m^2^ days 2–4, doxorubicin 40 mg/m^2^ day 1, cisplatin 40 mg/m^2^ days 3–4 + mitotane.

## Data Availability

The data presented in this study are available on request from the corresponding author. All figures presented in this article are original figures.

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
