# Peer review of "Feasibility and Activity of Megestrol Acetate in Addition to Etoposide, Doxorubicin, Cisplatin, and Mitotane as First-Line Therapy in Patients with Metastatic/Unresectable Adrenocortical Carcinoma with Low Performance Status"

_cancers, 2023, doi:10.3390/cancers15184491_

Round 1

Reviewer 1 Report

This is an interesting and well-written manuscript on the addition of megestrol-acetate to the EDP-M regimen. As expected, the addition of megestrol did not change the data of survival, but it can increase the tolerability of the regimen. The study is well-built and the results are not exaggerated. The language of the manuscript is fine and easy to follow. I propose the acceptance of the manuscript in its present form.

Author Response

We thank the Reviewer for the positive evaluation.

Reviewer 2 Report

The current manuscript reports the results of the study exploring the effects of megestrol acetate in association with EDP-M in adrenocortical cancer (ACC) patients with compromised PS with the aim to improve the patient’s general conditions and mitigate some side effects of EDP-M, demonstrating that the combination of EDP-M and megestrol acetate is feasible in ACC patients with low PS and it could allow the administration of the EDP-M regimen at a dose in tensity comparable to that administered to patients with good PS with simal efficacy (non-inferiority study). The study is interesting because ACC is a rare cancer with few treatment options for which EDP-M is 1st line treatment. Unfortunately, this aggressive therapeutic schema can be not well tolerated by some patients, therefore new strategy to increase patients’ tolerability might have great clinical impact. The manuscript is well written, and results are well described and discussed.

I have only few comments:

-Considering table 1 some important prognostic factors are missing in a considerable number of patients. Although the two groups of patients were comparable according with the propensity score matching methodology, authors should try to complete the missing values or discuss this point. Future randomized study will further clarify this issue.  

-This reviewer is just wondering if the use of a control group with similar PS would have produced clearer results in terms of tolerability of treatment.  

 -The inclusion criteria for the control group should be better describe (i.e. absence of other disease).

 -Since the response and the tolerability to chemotherapy can be influenced by the duration and dose of mitotane treatment and by the mitotane blood levels, these parameters should be clearly described and considered.

 -Since the tolerability to mitotane and chemotherapy in general can be influenced by the suboptimal glucocorticoid replacement treatment, the replacement status of the patients should be clearly described and considered.

 -Might the megestrol acetate have an effect glucocorticoid receptor? This could be interesting to discuss for the potential clinical implication in this type of patients.

 -Page 3, line 131: the word “secondary” must be removed.    

Author Response

Reviewer 2: “The current manuscript reports the results of the study exploring the effects of megestrol acetate in association with EDP-M in adrenocortical cancer (ACC) patients with compromised PS with the aim to improve the patient’s general conditions and mitigate some side effects of EDP-M, demonstrating that the combination of EDP-M and megestrol acetate is feasible in ACC patients with low PS and it could allow the administration of the EDP-M regimen at a dose intensity comparable to that administered to patients with good PS with simal efficacy (non-inferiority study). The study is interesting because ACC is a rare cancer with few treatment options for which EDP-M is 1st line treatment. Unfortunately, this aggressive therapeutic schema can be not well tolerated by some patients, therefore new strategy to increase patients’ tolerability might have great clinical impact. The manuscript is well written, and results are well described and discussed.”

Reply: We thank the reviewer for the positive evaluation and the constructive remarks.

Reviewer 2: “I have only few comments:

  1. Considering table 1 some important prognostic factors are missing in a considerable number of patients. Although the two groups of patients were comparable according with the propensity score matching methodology, authors should try to complete the missing values or discuss this point. Future randomized study will further clarify this issue.”

Reply: As noted correctly by the referee, some data on prognostic factors were missing in Table 1. Some missing info has been recovered, such as having done the adjuvant mitotane therapy, others unfortunately not, as the GRAS score at diagnosis. For clarity, we added the number of missing patients for each prognostic parameter.  We have added this limitation to the discussion.

  1. Reviewer 2: “This reviewer is just wondering if the use of a control group with similar PS would have produced clearer results in terms of tolerability of treatment.”

Reply: In principle the referee is right, the balanced groups for PS could provide more accurate information on tolerability and efficacy. Unfortunately, we do not have a control group with low PS as we decided in these patients to add progestin to improve patient cenesthesia, counteract weight loss, and increase appetite. It should be noted, however, that Performance Status has an important impact on treatment tolerability and generally we expect the tolerability to be worse in patients with low PS. Since the addition of megestrol acetate did not worsen the tolerability of the EDP-M regimen in patients with low PS, it could be assumed that the combination is feasible even in fit patients.

  1. Reviewer 2: “The inclusion criteria for the control group should be better describe (i.e. absence of other disease).”

Reply: The inclusion criteria for the control group were better pointed out in the paragraph “2.1. Patients” (page 3).

  1. Reviewer 2: “Since the response and the tolerability to chemotherapy can be influenced by the duration and dose of mitotane treatment and by the mitotane blood levels, these parameters should be clearly described and considered.”

Reply: The duration and dose of mitotane treatment and the mitotane blood levels were added in paragraph “3.2. Treatment-related toxicities”; no statistical difference was observed between the 2 groups.

  1. Reviewer 2: “Since the tolerability to mitotane and chemotherapy in general can be influenced by the suboptimal glucocorticoid replacement treatment, the replacement status of the patients should be clearly described and considered.”

Reply: The glucocorticoid replacement treatment was described in the paragraph “3.2. Treatment-related toxicities”; no statistical difference was observed.

  1. Reviewer 2: “Might the megestrol acetate have an effect glucocorticoid receptor? This could be interesting to discuss for the potential clinical implication in this type of patients.”

Reply: We thank the Reviewer for his/her observation. Indeed, it has been shown that megestrol acetate could bind glucocorticoid receptors in different experimental models, both normal and cancer tissues (PMID: 6248208; doi: 10.5966/sctm.2015-0009). Although data on this phenomenon are scarce, we agree that megestrol acetate could exert its effect by activating not only PgR, but also other steroid receptors (PMID: 6248208; DOI: 10.1016/0022-4731(88)90295-6).

Being able to settle the contribution of the binding and activation of receptors other than PgR in modulating the effect of megestrol acetate in patients remains however difficult to evaluate, as results reported in the above-cited articles have been obtained in in vitro models, at µM megestrol concentrations, that are not usually reached with the megestrol acetate dosage usually administered in the clinical practice for cachexia. Indeed, the plasmatic megestrol acetate concentration, even at the mean Cmax at a dosage as high as 800mg/die, results in about 1-3 µM concentration (Megestrol acetate In: MerativeTM Micromedex® DRUGDEX® (electronic version). Merative, Ann Arbor, Michigan, USA. Available at: https://www.micromedexsolutions.com/ (cited: August, 23, 2023).

However, we do agree with the Reviewer’s observation and this point has been now addressed in the paragraph “4. Discussion” (page 9).

  1. Reviewer 2: “Page 3, line 131: the word “secondary” must be removed. “

Reply: Page 3, line 131: the word “secondary” was removed. 

Reviewer 3 Report

Dear Author,

It is an interesting article that brings useful information related to the treatment of adrenal cancer.

The quality of the figures and tables is satisfactory.

The reference list covers the relevant literature adequately and impartially.

Statistical methods are valid and correctly applied.

In my opinion  it meets the conditions for publication.

Kind regards,

Author Response

We thank the Reviewer for the positive comments.
